# A Pathway for the Integration of Novel Ferroelectric Thin Films on Non-Planar Photonic Integrated Circuits

**DOI:** 10.3390/mi16030334

**Published:** 2025-03-13

**Authors:** Enes Lievens, Kobe De Geest, Ewout Picavet, Liesbet Van Landschoot, Henk Vrielinck, Gilles Freddy Feutmba, Hannes Rijckaert, Klaartje De Buysser, Dries Van Thourhout, Peter Bienstman, Jeroen Beeckman

**Affiliations:** 1LCP Group, Department of Electronics and Information Systems, Ghent University, Technologiepark-Zwijnaarde 126, 9052 Gent, Belgium; enes.lievens@ugent.be (E.L.); kobe.degeest@ugent.be (K.D.G.); ewout.picavet@ugent.be (E.P.); gillesfreddy.feutmba@ugent.be (G.F.F.); 2PRG Group, Department of Information Technology, Ghent University—imec, Technologiepark-Zwijnaarde 126, 9052 Ghent, Belgium; liesbet.vanlandschoot@ugent.be (L.V.L.); dries.vanthourhout@ugent.be (D.V.T.); peter.bienstman@ugent.be (P.B.); 3SCRiPTS, Department of Chemistry, Ghent University, Krijgslaan 281-S3, 9000 Ghent, Belgium; hannes.rijckaert@ugent.be (H.R.); klaartje.debuysser@ugent.be (K.D.B.); 4DiSC, Department of Solid State Sciences, Ghent University, Krijgslaan 281-S1, 9000 Ghent, Belgium; henk.vrielinck@ugent.be

**Keywords:** hydrogen silsesquioxane, planarization, ferroelectric thin films, heterogeneous integration, chemical solution deposition, Pockels modulator

## Abstract

The heterogeneous integration of ferroelectric thin films on silicon- or silicon nitride-based platforms for photonic integrated circuits plays a crucial role in the development of nanophotonic thin film modulators. For this purpose, an ultrathin seed film was recently introduced as an integration method for ferroelectric thin films such as BaTiO_3_ and Pb(Zr,Ti)O_3_. One issue with this self-orienting seed film is that for non-planarized circuits, it fails to act as a template film for the thin films. To circumvent this problem, we propose a method of planarization without the need for wafer-scale chemical mechanical polishing by using hydrogen silsesquioxane as a precursor to forming amorphous silica, in order to create an oxide cladding similar to the thermal oxide often present on silicon-based platforms. Additionally, this oxide cladding is compatible with the high annealing temperatures usually required for the deposition of these novel ferroelectric thin films (600–800 °C). The thickness of this silica film can be controlled through a dry etch process, giving rise to a versatile platform for integrating nanophotonic thin film modulators on a wider variety of substrates. Using this method, we successfully demonstrate a hybrid BaTiO_3_-Si ring modulator with a high Pockels coefficient of $r_{wg}=155.57\pm 10.91$ pm V^−1^ and a half-wave voltage-length product of $V_{\pi }L=2.638\pm 0.084$ V cm, confirming the integration of ferroelectric thin films on an initially non-planar substrate.

## 1. Introduction

The outstanding functional properties of certain ferroelectric materials have sparked significant interest in their heterogeneous integration into nanophotonic circuits, enabling a wide range of novel photonic devices [1,2,3,4,5]. In particular, silicon (Si) photonics can potentially benefit greatly from the incorporation of ferroelectric thin films, which are essential for developing the next generation of nanophotonic modulators. Recent advances in deposition techniques such as molecular beam epitaxy (MBE), pulsed laser deposition (PLD) and chemical solution deposition (CSD) have facilitated the growth of high-quality thin films with excellent crystallinity and interface control. However, the integration of these novel thin films on a silicon platform requires complicated techniques such as wafer bonding, transfer printing and flip chip [6,7,8,9]. Our approach is unique in the way that we can directly integrate electro-optic (EO) thin films on a silicon photonics platform, without the need to transfer the grown film over from another substrate [10,11,12]. These developments have led to significant improvements in EO modulation efficiency, reduced power consumption and expanded operational bandwidth, enhancing the performance of integrated photonic devices. Ferroelectric materials such as lithium niobate (LiNbO_3_) [13,14], barium titanate (BaTiO_3_) [11,15,16] and lead zirconate titanate (Pb(Zr,Ti)O_3_) [12,17,18] have been successfully integrated into photonic integrated circuits (PICs) to leverage their strong EO properties. These materials provide a reliable means of achieving phase modulation while mitigating spurious amplitude modulation, a challenge faced by traditional silicon modulators based on carrier dispersion. In these materials, the linear EO effect (also known as the Pockels effect) is described by a linear relation between the refractive index Δ*n* and the applied electric field *E*, given by the following simplified equation [11]:(1)$$\Delta n\left(E\right)=-\frac{1}{2}r_{\mathrm{eff}}n_{0}^{3}E$$

In this equation, Δ*n* represents the change in refractive index, *r_eff_* is the effective Pockels coefficient and *n*_0_ is the refractive index in the absence of an electric field. Despite the outstanding EO properties of these ferroelectric thin films, their integration remains far from trivial. For this reason, research on different integration routes of thin films has gained popularity in recent years [13,14,15,16,17,19,20,21]. Recently, a general integration method was introduced that allows for the heterogeneous integration of novel ferroelectric thin films on a wide variety of substrates [10]. This method relies on a chemical solution deposition (CSD) process, by spin coating an ultrathin La_2_O_2_CO_3_ template film. Since this seed film is very thin (∼8 nm), a relatively flat surface is required in order for it to act as a template film for ferroelectric thin film growth. When deposited on a non-planar substrate, no continuous microstructure is observed on the waveguide and trenches of the circuit, indicating that the ferroelectric film fails to grow properly, as is shown in Figure 1.

To tackle this problem, we propose a planarization method relying on the CSD of hydrogen silsesquioxane (HSQ), resulting in an oxide cladding closely resembling thermal oxides grown on silicon (Si)-based platforms [22,23]. This method circumvents the need for wafer-scale chemical mechanical polishing (CMP), as it is not always readily available in a research environment where there is a need for prototyping on a die level. In such an environment, PICs are typically fabricated on a single die through electron beam lithography [24,25,26]. This means that the substrate is too small to reliably use CMP. In order to quantify the planarization, two metrics are commonly introduced. First, the reduction in step height is evaluated through the degree of planarization $\gamma_{P}$, which is defined through the following equation [27]:(2)$$\gamma_{P}=1-\frac{\Delta_{coated}}{\Delta_{uncoated}}$$

In this equation, $\Delta_{coated}$ is the step height after coating, and $\Delta_{uncoated}$ is the step height of the uncoated substrate. Second, the reduction in step angle $\theta_{P}$ is evaluated. A visual representation of these parameters is provided in Figure 2. In the ideal case, $\gamma_{P}=1$ and $\theta_{P}=0$, which means the substrate is fully planar. However, in reality, the surface topology of the substrate will still be partially present after the spin coating of the HSQ film and $\gamma_{P}<1$ and $\theta_{P}>0$. The exact values of $\gamma_{P}$ and $\theta_{P}$ depend on several factors, such as the initial surface topology, film thickness and the rheological properties of the solvent, as well as the wettability of the substrate and the deposition method. Several experimental studies on the planarization properties of spin-on dielectrics have been published, and several important trends have been observed [27,28,29]. First, the degree of planarization increases as the density of features increases. Second, the degree of planarization increases as the thickness of the spin-on dielectric increases. Finally, the degree of planarization improves during an additional post-bake step.

In this work, the solution-deposited HSQ film is first annealed at high temperature in an oxygen-rich environment to form an amorphous silica layer, which is characterized using Fourier transform infrared (FTIR) spectroscopy. Second, reactive ion etching (RIE) is used to etch back the deposited silica layer to minimize the film thickness on top of the waveguide, as a smaller silica film on top of the waveguide will result in a higher EO mode overlap. Finally, a hybrid BaTiO_3_-Si ring modulator is fabricated in order to verify the proposed integration route, providing a low-cost, easy-to-implement planarization method. In this work, PICs fabricated on a 220 nm thick silicon-on-insulator (SOI) stack are planarized.

## 2. Materials and Methods

Hybrid BaTiO_3_-Si ring resonators were designed based on a 220 nm SOI stack. The design was patterned onto the SOI substrate using a Raith Voyager Electron Beam Lithography system (Raith GmbH, Dortmund, Germany). After lithography, an Oxford PlasmaPro 100 Cobra ICP RIE System (Oxford Instruments, Abingdon, UK) was used to etch the desired photonic circuit into the 220 nm thick silicon top layer of the SOI substrate. Next, the PICs were cleaned using a combination of oxygen plasma cleaning and UV–ozone cleaning. The oxygen plasma cleaning was conducted in a PVA TePla GIGAbatch 310M plasma system (PVA TePla AG, Wettenberg, Germany) for 20 min, followed by a 15 min UV–ozone clean using a Novascan PSD-UV Pro Series benchtop UV–ozone cleaner (Novascan Technologies, Inc., Ames, IA, USA). The cleaning was performed to rid the substrates of any surface contamination before spin coating HSQ. Afterwards, HSQ is deposited on the substrate by spin coating the commercially available product FOx-15 (Dow Corning Corporation, Midland, MI, USA) at 1500 rpm for 60 s. Here, a low spin speed is used in order to maximize film thickness, with 1500 rpm being the lowest reliable spin speed found without affecting film uniformity. Afterwards, the HSQ film is baked at 150 °C for one minute on a hotplate in ambient air before being annealed in a Jipelec jetfirst 150 rapid thermal annealing (RTA) furnace in 1000 sccm O_2_ for 30 min at 650 °C, using a heating rate of 60 °C/min. At these elevated temperatures, reactions occur, which convert the porous HSQ into a relatively dense amorphous silica. During these reactions, structural changes occur through which cagelike HSQ undergoes thermally activated bond rearrangement to form a three-dimensional network structure, resulting in a conversion of the Si-H bonds to Si-O bonds and a densification of the HSQ film, as well as a slight increase in the dielectric constant [30,31,32,33]. A thicker film is spin-coated as opposed to a thin film, as a thicker film will have better gap fill properties and therefore will be more suited to planarize the surface. In the context of this work, the ferroelectric thin film should be as close as possible to the waveguides, in order to maximize the EO overlap integral and therefore the modulation efficiency of fabricated devices. For this reason, the excess silica is etched away using an RIE system, resulting in a flat thin film with good gap fill properties. An Oxford PlasmaPro 100 Cobra ICP RIE System (Oxford Instruments, Abingdon, UK) is used to etch away the oxide using a combination of 200 sccm CHF_3_ and 5 sccm O_2_ with an RF power of 150 W and chamber pressure of 15 mTorr, resulting in a stable etch rate of the silica film of 16–18 nm/min. Using this etch recipe, the deposited oxide film was reliably etched down to an average film thickness of 20–40 nm above the waveguide without significantly affecting the PIC performance. After planarization, the seed film and BaTiO_3_ were deposited through a CSD process, resulting in a 120 nm thick BaTiO_3_ film as described in [10,11]. Finally, electrode patterns were deposited using UV lithography using a SUSS MicroTec MA6 Mask Aligner (SUSS MicroTec, Garching bei München, Germany), Ti/Au e-beam evaporation with a Leybold UNIVEX (Leybold GmbH, Cologne, Germany) and lift-off in acetone, resulting in a photonic chip with a cross-section as shown in Figure 3.

The DC operation of the hybrid BaTiO_3_-Si ring resonator was then investigated to confirm device operation using a benchtop measurement setup. A Santec TSL-510 tuneable semiconductor laser was used to generate light at a wavelength of 1550 nm. This light was then coupled into and out of the photonic chip via grating couplers designed for a 220 nm SOI PIC at an incident angle of 10°. The transmitted spectrum was recorded via a Newport 1936-R optical power meter. Finally, a DC field was applied across the ring with a Keithley 2400 (Keithley Instruments LLC, Cleveland, OH, USA) source measure unit through MPI coaxial DC probes (MPI Corporation, Hsinchu, Taiwan), which landed on the gold electrodes of the ring resonator.

## 3. Results

### 3.1. HSQ Film Characterization

The goal of this work is to introduce a planarization pathway in PICs without adversely affecting its performance by using HSQ as a precursor to forming amorphous silica, circumventing the need for wafer-scale CMP. Therefore, the properties of the deposited film should be characterized before and after processing. The deposited HSQ films were characterized by means of attenuated total reflectance (ATR) FTIR using ZnSe as the internal reflection medium (IRM). The resulting spectra are shown in Figure 4.

In Figure 4, the transformation of HSQ to amorphous silica is confirmed by ATR-FTIR. Before annealing, various Si-H and Si-O bands characteristic to HSQ are observed. Si-H bands are found between frequencies of 2080–2280 cm^−1^ and 800–950 cm^−1^. Si-O bands are found between 1060 and 1110 cm^−1^ for all forms of silica, with additional bands at frequencies of 780, 795, 800, 1170 and 1200 cm^−1^ for different phases of crystalline silica and 800–810 and 1220 cm^−1^ for amorphous silica. The presence of all Si-H bands and a mix of Si-O bands at 780–800 and 1060–1170 cm^−1^ is observed in the HSQ film before annealing [34]. After the annealing step, no more Si-H bands are observed, and three different Si-O bands remain at 800–810, 1060 and 1170 cm^−1^, with an absorption band shape that is almost identical to that of the thermally grown oxide. It confirms that annealing in an oxygen environment does indeed transform HSQ into amorphous silica closely resembling a thermal oxide.

### 3.2. Degree of Planarization

As for the planarization, cross-sectional SEM images were created by using a FEI Nova 600 Nanolab dual-beam FIB system (FEI Company, Hillsboro, OR, USA). These samples had an initial step height of 220 nm and trench width of 2 µm. An SEM cross-section is shown in Figure 5a. Initially, a step height of 220 nm is measured between the middle of the trench and top of the waveguide. After the deposition and annealing of HSQ, a clear improvement in surface flatness is observed. Using Equation (Equation 2), a degree of planarization of $\gamma_{P}=0.91$ is found. This corresponds to a 91% reduction in step height from the original 220 nm to a step height of 20 nm. Additionally, a step angle of $\theta_{P}=5.2$° was measured. Next, the oxide is etched down as close to the buried silicon structures as possible, without completely etching through the deposited oxide cladding. Finally, BaTiO_3_ is deposited on top of the planarized PIC, and SEM analysis is performed on the resulting stack in order to verify if the deposition and growth of the BaTiO_3_ film were successful. In Figure 5b, it can be seen that after etching, some planarization is lost, resulting in a degree of planarization of $\gamma_{P}=0.66$ and step angle of $\theta_{P}=31.7$°, leaving a step height of 75 nm. In Figure 5c, it is observed that the microstructure of the BaTiO_3_ has grown over the entire section, indicating that the final surface is still planar enough to allow for the growth of our novel ferroelectric thin films. Grains with a diameter of 30–90 nm are observed, resulting in an average grain size of roughly 60 nm. The dense, grain-like microstructure observed here is similar to that of the microstructure reported in [11]. The roughness of the BaTiO_3_ film was quantified by surface analysis using atomic force microscopy (AFM) and yielded a root mean square (RMS) roughness value of 3.00 nm (Appendix A). Additionally, X-ray diffraction (XRD) measurements of the BaTiO_3_ film were performed to confirm the preferred out-of-plane orientation caused by La_2_O_2_CO_3_ template film and are shown in Appendix A.

### 3.3. Cladding Thickness and Optical Mode Overlap

Before utilizing HSQ to assist in the integration of ferroelectric thin films, it is crucial to assess the impact of the film as a cladding material on the overall device performance. After deposition and annealing, the film is etched away until only a thin layer of oxide is left on top of the waveguide. As a rule of thumb, the thinner the cladding on top of the silicon waveguide, the higher the mode overlap will be, with a higher mode overlap resulting in more efficient modulation. The EO overlap integral Γ in the BaTiO_3_ film as a function of thickness off the silica film on top of the waveguide is shown in Figure 6. In order to increase the accuracy of the simulation, the surface of the annealed HSQ film was modeled to more precisely follow the shape of the film seen in the SEM images. It was found that the shape of this surface has a non-negligible effect on the value of the EO overlap integral, leading to overestimation of the Pockels coefficient of the BaTiO_3_ film when modeled as a perfectly flat surface. For a PIC with a top oxide of 20 nm thick, the EO overlap integral was found to be Γ=0.267. Here, a higher EO overlap value poses both an advantage and a disadvantage. In terms of modulation, a very high EO overlap value is favorable as this translates directly in higher modulation efficiency and therefore a lower half-wave voltage-length product *V_π_L*. However, a higher EO overlap usually means more light is in the BaTiO_3_ film, which can be a problem if the BaTiO_3_ film presents additional optical losses. Currently, most of the state-of-the-art BaTiO_3_ films still introduce unwanted optical losses, so higher EO overlap will usually lead to increased optical losses [35,36,37,38,39].

### 3.4. Integration of BaTiO_3_ into Silicon Ring Resonator

To validate successful integration, a BaTiO_3_ thin film was deposited onto a Si C-band ring resonator (see Materials and Methods section), and its DC operation was investigated (see Figure 7. Before the EO modulation, we first determined the ferroelectric properties of the BaTiO_3_ film (Appendix A). Additionally, the permittivity of the BaTiO_3_ film was characterized using in-plane electrical measurements using an impedance analyzer, resulting in a permittivity of 3960. Light with a wavelength centered around 1550 nm was coupled into the waveguide using grating couplers. The transmission of the hybrid BaTiO_3_-Si C-band ring resonator is shown in Figure 8a. The Pockels effect is verified by applying a DC field and measuring the shift in transmission spectrum. The applied DC field causes a change in the refractive index of the BaTiO_3_ film, which changes the optical path length in the ring, causing a shift in the resonant wavelength of the ring resonator [40]. The resulting shift of the resonance wavelengths as a function of the applied voltage is plotted in Figure 8b. From this, the half-wave voltage-length product, which is defined as(3)$$V_{\pi }L=\frac{L\lambda_{FSR}\Delta V}{2\Delta \lambda }$$
can be calculated. Here, *L* (523.6 µm) is the length of the modulated section of the ring, $\lambda_{FSR}$ (1.015 nm) is the free spectral range and Δλ/ΔV (10.07±0.32 pm/V) is the slope of the resonant shift as a function of the applied voltage of the ring, which is also called the tuning efficiency. Using Equation (Equation 3), a half-wave voltage-length product of *V_π_L*=2.638±0.084 Vcm is obtained. Although it is a promising first result on the integration of our in-house BaTiO_3_ on a SOI platform, the half-wave voltage-length product is not low enough to be considered state-of-the-art. However, many design considerations can be taken into account to improve upon this work. For example, in this work, electrodes were used with a spacing of 6 µm to achieve modulation. Decreasing the electrode spacing from 6 µm to 2 µm alone would be enough to obtain a half-wave voltage-length product below 1 Vcm. The optimization of the device stack is therefore planned as future work. From the half-wave voltage-length product, one can calculate the Pockels coefficient through the following equation [11]:(4)$$r_{wg}=\frac{\lambda g}{\Gamma n^{3}V_{\pi }L}$$

Here, λ (1550 nm) is the wavelength, *g* (6 µm) the spacing between the electrodes, *n* (2.1) the refractive index of the BaTiO_3_ film and Γ (0.267) the electro-optic overlap integral. Using these results in a Pockels coefficient of $r_{wg}=155.57\pm 10.91$ pm V^−1^. The error is due to small variations in processing conditions, such as top oxide thickness, BaTiO_3_ film thickness and electrode spacing. These results are comparable to those earlier reported for BaTiO_3_ films, confirming that we have indeed grown a high-quality BaTiO_3_ film with an excellent EO response [11]. This opens the possibility of integrating different EO thin films on non-planar surfaces through a CSD method of HSQ.

## 4. Conclusions

An easy-to-use, low-cost pathway is provided to aid in the integration of novel ferroelectric thin films on non-planar substrates through the chemical solution deposition of HSQ. A three-step process including deposition, anneal and etch is proposed to transform HSQ into a thin silica film with a chemical structure that resembles that of the thermal oxide present in silicon substrates. A degree of planarization of $\gamma_{P}=0.66$ and step angle of $\theta_{P}=31.7$° are obtained on a substrate with an initial step height of 220 nm, resulting in a final step height of 75 nm. Furthermore, to test the planarization pathway, a hybrid BaTiO_3_-Si ring modulator with a $V_{\pi }L=2.638\pm 0.084$ V cm and $r_{wg}=155.57\pm 10.91$ pm V^−1^ is fabricated, confirming sufficient planarization to create a functional nanophotonic Pockels modulator based on BaTiO_3_.

## Figures and Tables

**Figure 1 micromachines-16-00334-f001:**
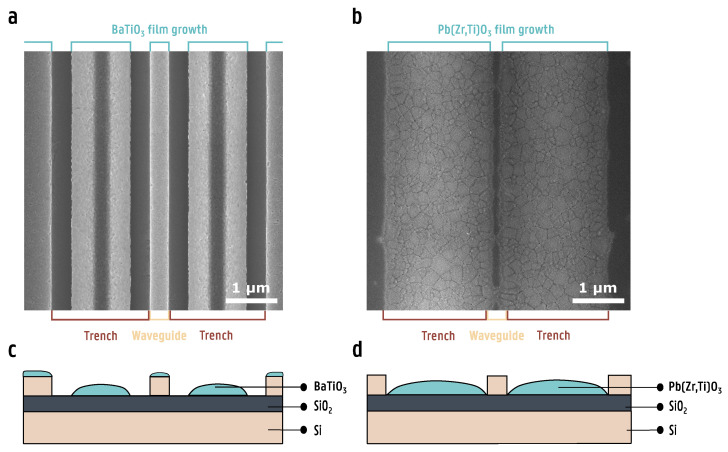
(**a**) Scanning electron microscopy (SEM) top view of BaTiO_3_ deposited on top of a non-planar silicon chip. (**b**) SEM top view of Pb(Zr,Ti)O_3_ deposited on top of a non-planar silicon waveguide. In both cases, the film does not grow homogeneously across the entire structure, with areas where no film has grown. (**c**) A cross-sectional illustration of a poorly grown BaTiO_3_ film. (**d**) A cross-sectional illustration of a poorly grown Pb(Zr,Ti)O_3_ film.

**Figure 2 micromachines-16-00334-f002:**
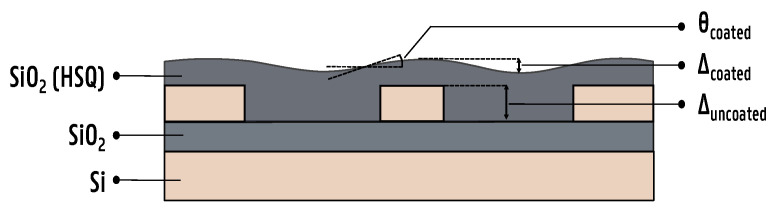
A cross-sectional illustration of the planarization process. HSQ acts as a suitable candidate for spin-on dielectrics with its relatively low dielectric constant and excellent gap fill properties. After annealing, the HSQ film undergoes structural changes and is transformed into amorphous silica. In this illustration, $\Delta_{\mathrm{uncoated}}$ is the height difference between the top of the waveguide and the bottom of the trench before planarization, $\Delta_{\mathrm{coated}}$ is the height difference between the oxide on top of the waveguide and the oxide in the trench after planarization and $\theta_{\mathrm{coated}}$ is the maximum step angle measured on the surface between the trench and waveguide.

**Figure 3 micromachines-16-00334-f003:**
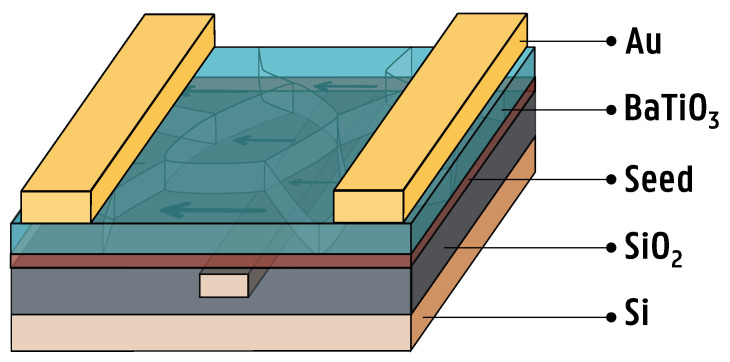
An illustration of the hybrid BaTiO_3_-Si waveguide stack. Starting from a 220 nm SOI substrate, 300 nm wide waveguides are etched into the silicon and planarized with amorphous silica. Next, a seed film and a 120 nm thick BaTiO_3_ film are deposited on top as described in [10,11]. Finally, 400 nm thick gold electrodes with a spacing of 6 µm are deposited on top of the BaTiO_3_ to allow for EO modulation.

**Figure 4 micromachines-16-00334-f004:**
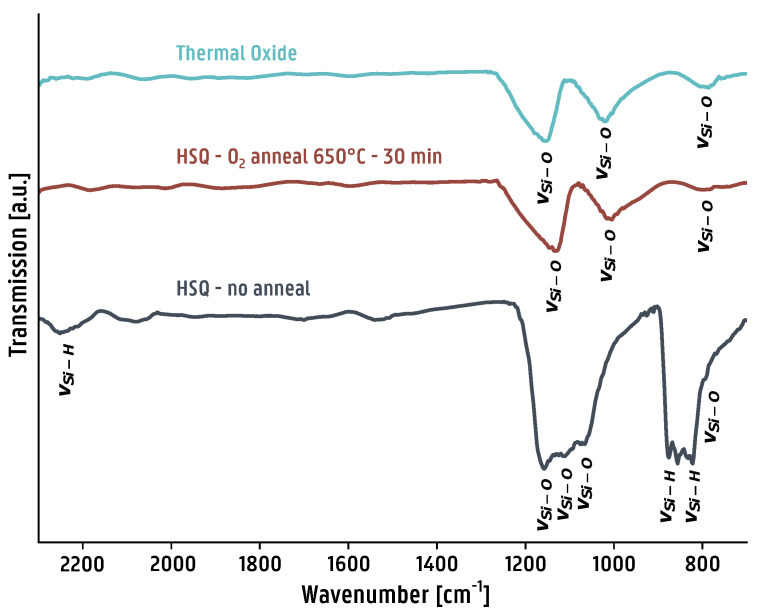
ATR-FTIR spectra of three different layers on top of a silicon wafer. The spectra of HSQ and HSQ annealed in oxygen are compared with the spectrum of a 500 nm thick thermally grown oxide on a silicon wafer in order to gain insight into the chemical structure of the films. The infrared absorption bands are identified based on a comparison with the literature [34].

**Figure 5 micromachines-16-00334-f005:**
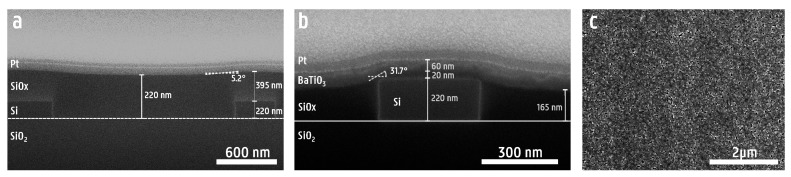
(**a**) An SEM cross-section of a 220 nm SOI photonic chip, planarized with HSQ. No etching has been performed yet. (**b**) An SEM cross-section of a 220 nm SOI photonic chip planarized with HSQ, after etching and BaTiO_3_ deposition. (**c**) An SEM top view of the dense BaTiO_3_ film successfully grown on a waveguide structure.

**Figure 6 micromachines-16-00334-f006:**
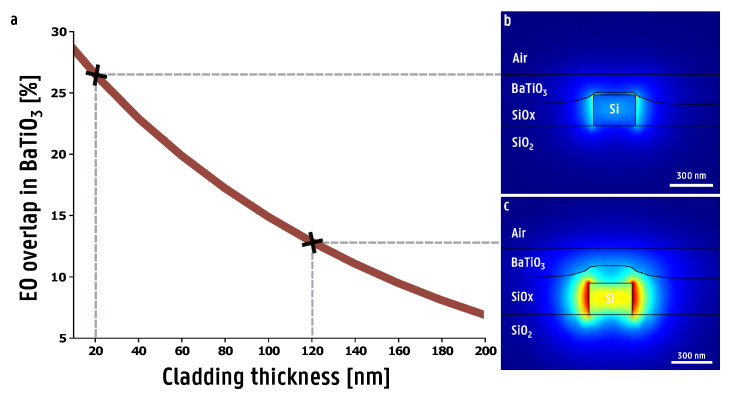
(**a**) The percentage of light of the guided mode confined within a 120 nm thick BaTiO_3_ film as a function of the silica film thickness on top of the silicon waveguide. The silicon waveguide has a width of 300 nm and a height of 220 nm. (**b**) The optical mode profile with a 20 nm thick silica film on top of the waveguide. For this configuration, an EO overlap integral of Γ=0.267 was calculated. (**c**) The optical mode profile with a 120 nm thick silica film on top of the waveguide. For this configuration, an EO overlap integral of Γ=0.131 was calculated.

**Figure 7 micromachines-16-00334-f007:**
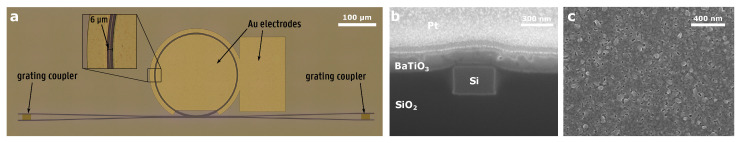
(**a**) A microscope image of the hybrid BaTiO_3_/Si ring modulator. The inset is a magnified image of a ring section with electrodes on top. The electrodes have a spacing of 6 µm. (**b**) An SEM cross-section after etching and BaTiO_3_ deposition. (**c**) A close-up SEM top view of the as-grown BaTiO_3_ top layer. A dense microstructure with small pores similar to the BaTiO_3_ reported in [11] is observed.

**Figure 8 micromachines-16-00334-f008:**
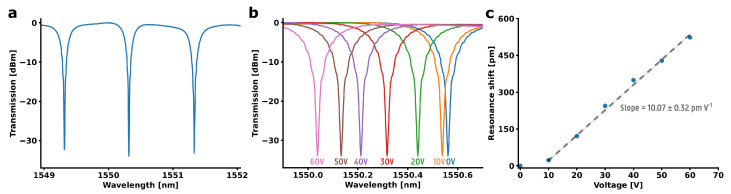
The transmission spectrum and resonance wavelength shift of a hybrid BaTiO_3_-Si C-band ring resonator. (**a**) The normalized transmission spectrum of the C-band ring resonator. (**b**) The normalized transmission spectra for different applied voltages. (**c**) The resonance wavelength shift as a function of the applied voltage.

## Data Availability

All data relevant to this research are provided in the main text.

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
