# Peer review of "A Pathway for the Integration of Novel Ferroelectric Thin Films on Non-Planar Photonic Integrated Circuits"

_micromachines, 2025, doi:10.3390/mi16030334_

Round 1

Reviewer 1 Report

Comments and Suggestions for Authors

The manuscript describes the integration of ferroelectric thin films on silicon waveguides for electro optic modulation. Experiments are well detailed, encompassing SEM cross sections for fabrication steps (HSQ spin and bake, HSQ thinning, BaTiO3 deposition), simulations, and experimental data on a ring resonator. The experiments are complete and well-described, and the motivation (low cost method avoiding CMP) and results are well described. While the method of using HSQ for planarization is not novel, the integration method of BaTiO3 on waveguides using HSQ and a seed layer is novel, and will be of interest to the community. 

I have some minor suggestions:

  • Figure 5, 7 : the thickness of the BaTiO3 seems to be not uniform at the spatial scale of the waveguide. Does this pose issues for the uniformity of performance of different waveguides? Also, does this cause high losses since a large part of the mode overlaps with the BaTiO3 thin film? The authors could either cite work or provide some additional details.
  • Figure 7: gold electrodes look shorted. Is this because of the small gap (6 microns)? It would be better to show a magnified picture of the electrode gap or mention the electrode separation in the figure legend
  • Line 39-41: language is hard to understand. Although there is a reference, it would be useful to mention that inject printing method is used as seed layer (this helps understand quickly why topography is an issue)
  • Figure 6: is it possible (and is it useful) to decrease further the thickness of the oxide on top of the waveguide to have a higher EO overlap in BaTiO3? What is the tradeoff between mode overlap and losses?
  • Figure 6 and 7 are inverted

Author Response

Comment 1: Figure 5, 7 : the thickness of the BaTiO3 seems to be not uniform at the spatial scale of the waveguide. Does this pose issues for the uniformity of performance of different waveguides? Also, does this cause high losses since a large part of the mode overlaps with the BaTiO3 thin film? The authors could either cite work or provide some additional details.

Thanks for this remark, it is indeed the case that the BaTiO3 thickness has some variation across the waveguide, due to the change in topography. We suspect that the performance will not vary significantly between different waveguides. This is because the Pockels coefficient is not dependent on the film thickness. Only the electro-optic overlap will vary slightly between different waveguides. This is only a minor variation between different waveguides, as this electro-optic overlap will not change significantly for small variations in thickness (see Fig. 6 in de original manuscript for an idea on the variation of this EO overlap). However, these minor variations are difficult to fully remove as both BaTiO3 and HSQ are spin coated, so there will always be slight variations in film thicknesses across a chip. 

As for the mode overlap in BaTiO3, this is indeed a good remark as in most recent works, the losses in BaTiO3 are not negligible. However, overlap in the BaTiO3 film is necessary to achieve modulation. We added additional details as well as references on BaTiO3 optical losses at the end of section 3.3, at page 6.

Comment 2: Figure 7: gold electrodes look shorted. Is this because of the small gap (6 microns)? It would be better to show a magnified picture of the electrode gap or mention the electrode separation in the figure legend

Thank you for pointing this out. A magnified picture is added as an inset of figure 7 with the electrode separation shown. The electrode gap is indeed 6 micron.

Comment 3: Line 39-41: language is hard to understand. Although there is a reference, it would be useful to mention that inject printing method is used as seed layer (this helps understand quickly why topography is an issue)

We have rewritten lines 39-41 to make it more understandable. The reference indeed mentions the use of inkjet printing, but this is not the method used for depositing the seed layer in this work. It is the same seed layer mentioned in the reference but spin coated instead. We found that spin coating gave more reproducible results, and allowed for easier scaling to bigger sample sizes. 

Comment 4: Figure 6: is it possible (and is it useful) to decrease further the thickness of the oxide on top of the waveguide to have a higher EO overlap in BaTiO3? What is the tradeoff between mode overlap and losses?

While it is theoretically possible to decrease the top oxide thickness even further, different tests showed that this gave issues both with performance and reproducibility. For a top oxide thickness of 20 nm, the EO overlap is found to be 26.7%,  for no top oxide, an EO overlap of about 29% is obtained. Given the same Pockels coefficient, this would decrease the VpiL from 2.638 Vcm to 2.225 Vcm. While not negligible, this does not drastically improve the performance in our opinion. 

Some issues arise when further decreasing the thickness. The top oxide is etched away at a rate of roughly 18 nm/min (conditions vary slightly over time, between 16-20 nm/min). This limits the resolution of how close we can reliably etch down. Additionally, local variations in top oxide thickness due to spin coating make it difficult to control the exact film thickness at every waveguide. We found that it is reliably possible to etch down to 20-30 nm oxide on top of the waveguide, without adversely affecting performance. Trying to decrease this thickness further often resulted in over etching and damage to the underlying waveguides.

For this reason we advise other approaches to have a higher EO overlap in BaTiO3. The three main ways are: (a) Decreasing electrode spacing, (b) Increasing BaTiO3 film thickness and (c) decreasing waveguide width. 

Presumably, using methods such as chemical mechanical polishing, it would be possible to reliably decrease the thickness to only a few nanometers, but is outside the scope of this work.

Finally, the main tradeoff between mode overlap and optical losses would be the increased losses currently reported in BaTiO3 films (such as the ones mentioned in comment 1). Creating ultra low-loss hybrid BaTiO3/Si (or BaTiO3/SiN) waveguides would therefore be possible only when low-loss BaTiO3 films can be manufactured.

A brief discussion of this is added at the end of section 3.3 on page 6.

Comment 5: Figure 6 and 7 are inverted

Thank you for pointing this out, there must have been a mistake in the LaTeX code and this has been corrected.

Reviewer 2 Report

Comments and Suggestions for Authors

The authors propose a method of planarization that eliminates the need for wafer-scale chemical mechanical polishing. This approach utilizes hydrogen silsesquioxane as a precursor to form amorphous silica, aiming to create an oxide cladding akin to the thermal oxide typically found on silicon-based platforms. I think this manuscript is well-organized and easy to follow. However, I recommend the author pay attention to the following comments to attract a broad interest among the readers of Micromachines.

  1. There is no crystal structure and chemical composition information. Hence, please add the XRD. Also, I recommend the author do the EDS or XPS measurements to support the crystal structure.
  2. How about the effects of thermal annealing time and temperatures on the film quality?
  3. Do the pores (Figure 7c) influence the properties? Please comment on this point.
  4. As pointed out in the title "novel ferroelectric thin film", thus please add the result or discussion on the intrinsic ferroelectrical properties.  
  5. In addition, I recommend the author add the AFM image to observe the surface morphology.
  6. Besides, I recommend the author add the recent progress in the heterogeneous integration.  

Author Response

Comment 1: There is no crystal structure and chemical composition information. Hence, please add the XRD. Also, I recommend the author do the EDS or XPS measurements to support the crystal structure.

It is indeed of added value to include some material characteristics. Therefore, XRD (θ-2θ) data was included in the SI (Figure S2) which verifies both the formation of the desired crystal structure and the formation of highly textured BaTiO3 films. EDS and XPS analysis is not included as elemental analysis such as EDX-XPS on thin films is very challenging because of the limited resolution. In addition, the Ba (Kα) & Ti (Kα) X-ray energies are similar which hampers appropriate data analyzation & interpretation. A thorough material analysis of the BaTiO3 film has been presented in the work of E. Picavet et al (https://doi.org/10.1002/adfm.202403024) which shows the crystal structure & chemical composition.

Comment 2: How about the effects of thermal annealing time and temperatures on the film quality?

For the BaTiO3 film, the thermal annealing time & temperature was optimized to obtain highly textured BaTiO3 films. Changing the thermal processing parameters such as thermal annealing time & temperature during BaTiO3 film growth will reduce the film texture & quality.

For the HSQ film, no significant changes were observed for annealing times longer than 50 minutes (10 min ramp up, 30 min hold, 10 min cooldown), and no significant changes were observed for annealing at temperatures higher than 650°C. So this film growth seems stable as long as it is done at a high enough temperature in oxygen. The lower limits of annealing time & temperature were not optimized further beyond 650°C and 50 minutes.

Comment 3: Do the pores (Figure 7c) influence the properties? Please comment on this point.

The effect of the pores is not yet fully understood, but it is reasonable to expect that the presence of these pores leads to an overall degradation of film quality an therefore EO effect. However, good EO properties are obtained even with a film that is not fully uniform.

Additionally, these pores could increase optical losses because they could act as scattering centers for propagating light. We do see an increase in propagation losses, but the exact origin of this increase is still being investigated within our group.

Comment 4: As pointed out in the title "novel ferroelectric thin film", thus please add the result or discussion on the intrinsic ferro electrical properties.  

Thank you for this addition, P-E measurements of the BaTiO3 films were performed and included in the SI (Figure S3). A narrow hysteresis with low remnant polarization was observed.

Comment 5: In addition, I recommend the author add the AFM image to observe the surface morphology.

An AFM image of the BaTiO3 film deposited on the cured HSQ is included in the SI (Figure S1), as it indeed adds value to better understand the film characteristics. 

Comment 6: Besides, I recommend the author add the recent progress in the heterogeneous integration.

The introduction was expanded slightly to incorporate some recent progress in heterogeneous integration, so that this work can be better situated in the field.

Round 2

Reviewer 2 Report

Comments and Suggestions for Authors

I think the authors make sufficient revisions. Thus, I suggest accept as the current version.